# Ganoderic Acid Ameliorates Ulcerative Colitis by Improving Intestinal Barrier Function via Gut Microbiota Modulation

**DOI:** 10.3390/ijms26062466

**Published:** 2025-03-10

**Authors:** Yuwei Ye, Abudumijiti Abulizi, Yukun Zhang, Feng Lu, Yongpan An, Chaoqun Ren, Hang Zhang, Yiming Wang, Dongmei Lin, Dan Lu, Min Li, Baoxue Yang

**Affiliations:** 1State Key Laboratory of Vascular Homeostasis and Remodeling, Department of Pharmacology, School of Basic Medical Sciences, Peking University, Beijing 100191, China; yuweiye1995@163.com (Y.Y.); abulizi@shzu.edu.cn (A.A.); yukun_zhang@bjmu.edu.cn (Y.Z.); fenglu202110@163.com (F.L.); anyongpan123@163.com (Y.A.); renchaoqun@pku.edu.cn (C.R.); hangzhang@bjmu.edu.cn (H.Z.); wangyiming931211@163.com (Y.W.); 2JUNCAO Technology Research Institute, Fujian Agriculture and Forestry University, Fuzhou 350002, China; lindm_juncao@163.com; 3Institute of Systems Biomedicine, Department of Pathology, School of Basic Medical Sciences, Peking University, Beijing 100191, China; taotao@bjmu.edu.cn

**Keywords:** ganoderic acid, ulcerative colitis, intestinal barrier function, gut microbiota, tight junction

## Abstract

Ulcerative colitis (UC) is a chronic and recurrent gastrointestinal disease that affects millions of humans worldwide and imposes a huge social and economic burden. It is necessary to find safe and efficient drugs for preventing and treating UC. The aim of this study was to determine whether ganoderic acid (GA), the main bioactive components of Ganoderma lucidum, has preventive and therapeutic effect on UC in a dextran sulfate sodium (DSS)-induced UC mouse model. Our experimental results showed that GA significantly ameliorated the body weight loss and disease activity index (DAI) of UC mice. GA significantly restored 11% of the colon length and 69% of the spleen index compared to UC mice. GA significantly decreased the intestinal inflammatory response and improved the barrier function of the intestine by upregulating the tight junction proteins Zonula occludens-1 (ZO-1), occludin and claudin-1. A co-housing experiment showed that gut microbiota accounted for the therapeutic activity of GA on UC, which was confirmed by fecal microbiota transplantation from GA-treated mice to the UC mice. Furthermore, 16S rDNA high-throughput sequencing of fecal bacteria showed that GA significantly enriched the abundance of Lactobacillus, Oscillospira, Odoribacter and Ruminococcus, which were positively correlated with colon length. Furthermore, this study found the functional metabolites, including Indole-3-acetaldehyde (IAAld), Glutamine (Gln) and Glutathione (GSH), reduced barrier damage in the Caco-2 cell model. In conclusion, this study suggests that GA could ameliorate UC by improving intestinal barrier function via modulating gut microbiota and associated metabolites.

## 1. Introduction

Ulcerative colitis (UC) is a chronic and recurrent gastrointestinal disease characterized by clinical symptoms such as diarrhea, abdominal pain, hematochezia, weight loss and fatigue [1]. Affecting millions of people worldwide, UC requires lifelong therapy, imposing a significant economic and social burden on patients and healthcare systems [2].

Despite extensive research, the etiology and pathogenesis of UC are still not fully understood. Current evidence suggests that UC arises from a complex interplay of genetic susceptibility, environmental factors, intestinal barrier dysfunction and gut microbial dysbiosis [3,4]. Environmental triggers, including diet, smoking, infections and antibiotic use, can disrupt gut microbiota homeostasis and compromise the intestinal barrier [5,6]. This disruption allows bacterial products to penetrate the intestinal wall, triggering an aberrant immune response characterized by the overproduction of pro-inflammatory mediators, the infiltration of immune cells and progressive tissue damage. Additionally, gut microbiota have emerged as a critical link between the external environment and the intestinal mucosa, playing a pivotal role in the pathogenesis of UC [7,8]. Studies have shown that UC patients exhibit significantly altered gut microbiota composition, with reduced species diversity and an imbalance in microbial metabolites, which further exacerbate intestinal inflammation [9,10,11]. Therefore, targeting these pathogenic processes, particularly gut microbiota modulation and intestinal barrier restoration, represents a promising therapeutic strategy for UC.

Classical treatments for UC include aminosalicylates, immunomodulators, corticosteroids and anti-cytokine drugs, which have a variety of adverse limitations [12]. In recent years, natural products derived from plants have gained attention for their therapeutic potential in UC, particularly due to their ability to modulate gut microbiota and restore intestinal barrier function with minimal side effects [13,14,15]. Among these natural compounds, ganoderic acid (GA), a major bioactive component of Ganoderma lucidum (G. lucidum), has emerged as a promising candidate.

Ganoderma lucidum, first recorded in Shennong’s Classic of Materia Medica, has been used for over 2000 years. Officially recognized in the Chinese Pharmacopoeia, this medicinal mushroom is rich in bioactive compounds, including polysaccharides and triterpenoids, and has been widely utilized in the treatment of various ailments such as nonalcoholic fatty liver disease, rheumatic arthritis, inflammation, and colon tumorigenesis [16,17,18]. GA, a triterpenoid compound, exhibits a wide range of pharmacological activities, including anti-oxidation, anti-inflammation, immune regulation, and hepatoprotection [19,20,21]. Recent studies have demonstrated that GA can mitigate inflammatory bowel disease and potentially ameliorate gut microbiota imbalance [22,23,24], leading us to speculate whether GA might prevent and treat UC via regulating gut microbiota.

In this study, we utilized a DSS-induced UC mouse model to investigate whether GA exerts its ameliorative effects on UC by modulating gut microbiota, as well as to explore its underlying mechanisms. Our studies aim to provide a theoretical foundation for future clinical drug development and therapeutic strategies.

## 2. Results

### 2.1. GA Ameliorated DSS-Induced UC in Mice

The DSS-induced UC mouse model was established as scheduled in Figure 1A. In this study, we administered three doses of GA to UC mice (DSSVeh group) at doses of 16.5 mg/kg, 50 mg/kg and 150 mg/kg, and sulfasalazine (SASP) was used as a control therapy at a dose of 150 mg/kg. DSSVeh mice showed a lower body weight and higher DAI than Veh mice (Figure 1B,C), indicating the successful establishment of the UC model. Interestingly, GA significantly ameliorated the body weight loss and DAI of DSSVeh mice, whose therapeutic effect was stronger than SASP at the same dose.

DSSVeh mice exhibited 43% shorter colon lengths and 112% higher spleen indexes than Veh mice, which were significantly restored by 11% and 69% by GA respectively (Figure 1D–F). Simultaneously, histopathological examination revealed the severe damage of the surface epithelium, destruction of the cryptal glands and infiltration of inflammatory cells in the mucosa of DSSVeh mice, which were improved by GA, suggesting that GA could protect against the disruption of colonic structures and reduce inflammatory infiltration in the mucosa (Figure 1G).

To further evaluate the effect of GA on intestinal inflammatory responses, the myeloperoxidase (MPO) and pro-inflammatory cytokines including interleukin-1β (IL-1β), interleukin-6 (IL-6), tumor necrosis factor-α(TNF-α) and anti-inflammatory cytokine interleukin-10 (IL-10) were detected. MPO activity was dramatically increased in the colonic tissue of DSSVeh mice, which was reversed by GA (Figure 1H). Accordingly, the mRNA and protein expression levels of IL-1β, IL-6 and TNF-α were significantly elevated, while the level of IL-10 was decreased in the colonic tissue of DSSVeh mice (Appendix A and Figure 1I–L). As expected, GA could effectively reduce the intestinal inflammatory response dose-dependently. In addition, the therapeutic experiments were performed by giving GA for 7 days since the third day after the mice were exposed to DSS. The experimental results showed that GA had a significant therapeutic effect on UC (Appendix A). These results indicate that GA ameliorates pathological symptoms and colonic inflammation in DSS-induced UC mice.

### 2.2. GA Improved Intestinal Barrier Function in DSS-Induced UC Mice

The barrier function of the intestine is closely related to the development of UC, and Clinical UC patients are usually associated with intestinal barrier disruption [25]. In order to determine the protective effect of GA on intestinal barrier function, we used FITC-dextran (FD4) as a tracer to evaluate the intestinal permeability. The serum FD4 levels of DSSVeh mice were much higher than those of Veh mice, which were significantly reduced by GA (Figure 2A).

The tight junction proteins Zonula occludens-1 (ZO-1), occludin, and claudin-1 play a key role in maintaining intestinal epithelial barrier integrity and cellular permeability [26,27]. Western blotting showed that the protein expression levels of ZO-1, occludin, and claudin-1 were obviously decreased in colonic tissues of DSSVeh mice compared with Veh mice, which were reversed by GA (Figure 2B,C). Accordingly, immunofluorescence staining further confirmed that ZO-1, occludin and claudin-1 proteins were severely deficient in the colonic tissue of DSSVeh mice, while GA could effectively maintain the expression of tight junction proteins (Figure 2D). These results indicate that GA could protect the intestinal barrier integrity and reduce intestinal permeability.

### 2.3. Gut Microbiota Were Involved in the Protective Effect GA on UC

To speculate whether gut microbiota are involved in the protective effect of GA on UC, we established a co-housing experiment for the transmissible nature of gut microbiota, as described in Figure 3A. As expected, the Co-DSSVeh mice a showed higher body weight and lower DAI score than the DSSVeh mice. Interestingly, Co-DSSGA mice had a lower body weight and higher DAI score than DSSGA mice (Figure 3B,C). In addition, Co-DSSVeh mice exhibited a longer colon length and lower spleen index compared with DSSVeh mice (Figure 3D and Appendix A). We also investigated the barrier function changes among different groups by detecting the protein expression levels of ZO-1, occludin and claudin-1. The results showed that these tight junction proteins’ expression was higher in the colonic tissues of Co-DSSVeh mice than in that of DSSVeh mice (Figure 3E,F). Simultaneously, histopathological examination revealed that co-housing treatment improved the pathological changes including the incomplete surface epithelium, the destruction of the cryptal glands and the infiltration of inflammatory cells in the mucosa in UC mice (Figure 3G).

Furthermore, colonic MPO activity and pro-inflammatory cytokines including IL-1β, IL-6 and TNF-α were decreased, while the anti-inflammatory cytokine IL-10 was elevated in DSSVeh mice as a result of the co-housing treatment (Figure 3H–L and Appendix A). Altogether, these results indicated that gut microbiota might have been exchanged between the Co-DSSGA mice and Co-DSSVeh mice, subsequently affecting the pathological symptoms and colonic inflammation of DSSVeh mice.

### 2.4. GA Alleviates Gut Microbiota Dysbiosis in DSS-Induced UC Mice

Previous studies have shown that gut microbiota dysbiosis plays a critical role in the development of UC [9,13,28]. 16S rDNA high-throughput sequencing of fecal bacteria was performed to investigate the impact of GA on the gut microbiota composition. The Principal co-ordinate analysis (PCoA) showed that the Veh group and GA group clustered together and were far away from the DSSVeh group, indicating that GA treatment did not cause significant changes in the gut microbiota of healthy mice (Figure 4A). In addition, the distance between the DSSGA group and Veh group was much closer than that between the DSSVeh group and Veh group, suggesting the gut microbial composition of DSSGA mice was more similar to that of Veh mice. The Gini–Simpson index, reflecting the α diversity of microbial composition, was higher in DSSVeh mice than that in Veh mice, which was decreased by GA treatment (Figure 4B).

At the phylum level, the main phyla were Firmicutes, Bacteroidetes and Proteobacteria (Figure 4C). Compared with the Veh mice, the relative abundance of Firmicutes was significantly decreased and Proteobacteria was increased in DSSVeh mice, while GA treatment could restore the changes of gut microbiota (Figure 4D,E). Linear discriminant analysis effect size (LEfSe) analysis was performed to further identify significantly altered bacterial taxa at the genus level between DSSVeh mice and DSSGA mice (Figure 4F). Compared with DSSVeh mice, the relative abundance of Lactobacillus, Oscillospira, Odoribacter and Ruminococcus was enriched in DSSGA mice (Figure 4G–J). Furthermore, the relationships between the top 10 abundant genera and colon length, MPO, DAI, spleen index and pro-inflammatory cytokines were analyzed to investigate whether the changes in gut microbiota in DSSVeh mice were correlated with intestinal inflammatory symptoms (Figure 4K). The abundance of Escherichia and Bacteroides, which were higher in DSSVeh mice, both exhibited a positive correlation with the MPO, DAI, spleen index and pro-inflammatory cytokines and exhibited a negative correlation with colon length. Inversely, the abundance of Lactobacillus, Oscillospira, Odoribacter and Ruminococcus, which were enriched in DSSGA mice, exhibited a negative correlation with the MPO, DAI, spleen index and pro-inflammatory cytokines and exhibited a positive correlation with colon length. The results suggest that GA could reverse gut microbiota dysbiosis associated with UC.

### 2.5. GA Ameliorated UC via Regulating the Gut Microbiota

To investigate the therapeutic potential of the gut microbiota affected by GA in the development of UC, we designed the Fecal microbiota transplantation (FMT) experiment (Figure 5A). As shown in Figure 5B,C, the FMT from the GA-treated donor mice significantly ameliorated weight loss and decreased the DAI score in recipient mice (FMT-DSSGA group). Compared with the FMT-DSSVeh mice, FMT-DSSGA mice exhibited a longer colon length and lower spleen index (Figure 5D and Appendix A). The protein expression levels of ZO-1, occludin and claudin-1 were higher in the colonic tissues of FMT-DSSGA mice than those of FMT-DSSVeh mice (Figure 5E,F). In addition, FMT-DSSGA mice exhibited milder intestinal pathological damage including an incomplete surface epithelium, the destruction of the cryptal glands and the infiltration of inflammatory cells in colonic mucosa (Figure 5G). Moreover, colonic inflammation was improved by FMT from GA-treated donor mice through determining the MPO activity, IL-1β, IL-6, TNF-α and IL-10 (Figure 5H–L and Appendix A). Collectively, these results indicate that GA exerts therapeutic potential for UC by regulating the gut microbiota.

### 2.6. GA Ameliorated UC by Gut Microbiota-Related Metabolites

There is growing evidence that gut microbiota-related metabolites could influence the pathogenesis of UC [10,11]. In order to explore the specific mechanism of gut microbiota affected by GA improving UC, we used an untargeted metabolomics approach to identify the differential metabolites in the colonic contents between DSSVeh mice and DSSGA mice. The partial least squares discrimination analysis (PLS-DA) showed significant metabolic differentiation between DSSVeh mice and DSSGA mice (Figure 6A). A total of 187 metabolites were identified between the two groups (Figure 6B). Pathway enrichment analysis showed that these differential metabolites affected by GA were involved in glycine, serine and threonine metabolism, bile secretion, tryptophan metabolism, cysteine and methionine metabolism, pyrimidine metabolism, vitamin digestion and absorption, vitamin B6 metabolism and prostate cancer based on the Kyoto Encyclopedia of Genes and Genomes (KEGG) database (Figure 6C). To further search for the metabolites that play a major role in the improvement of UC, Spearman’s correlation analysis was conducted between differential metabolites with a VIP > 3 and notable changed bacteria affected by GA. The Indole-3-acetaldehyde (IAAld), Glutamine (Gln) and Glutathione (GSH) levels were positively correlated with the abundance of Lactobacillus, Oscillospira, Odoribacter and Ruminococcus, which are the dominant genera under the effect of GA. Inversely, the Deoxycholic acid (DCA) level showed a negative correlation with the abundance of Lactobacillus, Oscillospira, Odoribacter and Ruminococcus (Figure 6D). In addition, the relative levels of IAAld, Gln and GSH were much higher and that of DCA was much lower in DSSGA mice than in DSSVeh mice (Figure 6E–I). These results indicate that gut microbiota-related metabolites affected by GA are involved in the development of UC.

### 2.7. IAAld, Gln and GSH Directly Protected Intestinal Barrier Damage

To further investigate the effect of the GA-influenced metabolites on intestinal barrier, we constructed a model of the intestinal epithelial barrier using Caco-2 and NCM460 cells, which form the polarized monolayers that resemble the intestinal epithelium [29]. In this study, we found IAAld, Gln and GSH could restore the reduction in trans-epithelial electrical resistance (TEER) caused by treatment with pro-inflammatory cytokines, which reflects the integrity of the barrier (Figure 7A,B). Moreover, IAAld, Gln and GSH also significantly attenuated the increase in the paracellular permeability of the cell monolayer, which was measured by Lucifer yellow (Figure 7C,D). Interestingly, DCA, which was negatively correlated with GA administration, reduced the TEER and enhanced the paracellular permeability of cell monolayers (Figure 7A–D). In addition, we found that IAAld, Gln and GSH could upregulate the expression of tight junction proteins and DCA could downregulate the expression of tight junction proteins of Caco-2 and NCM460 cells in the barrier damage model (Figure 7E–H). Overall, gut microbiota-related metabolites upregulated by GA had a direct protective effect on intestinal barrier function.

## 3. Discussion

Ulcerative colitis (UC) as a chronic and recurrent gastrointestinal disease that affects millions of people worldwide, and its pathogenesis involves various factors [2,3,4]. Natural products have garnered significant attention as potential therapeutic agents for UC due to their multi-targeted effects and low toxicity [14,30,31,32]. Among these, Ganoderic acid (GA), a major bioactive component of Ganoderma lucidum (G. lucidum), has demonstrated a wide range of pharmacological activities, including anti-oxidation, anti-inflammation, immune regulation and hepatoprotection [19,20,21]. In this study, we explored the therapeutic potential of GA in UC, focusing on its effects on colonic inflammation, intestinal barrier function and gut microbiota modulation.

Colonic inflammation of the gastrointestinal tract is a hallmark of UC [33]. Previous studies have showed a marked imbalance between pro-inflammatory and anti-inflammatory cytokines in UC patients and experimental models [14,34]. Consistent with previous studies, we found a significant increase in pro-inflammatory cytokines (TNF-α, IL-1β and IL-6) and a decrease in the anti-inflammatory cytokine IL-10 in the colonic tissue of UC mice. Notably, GA reversed these changes, suggesting its potent anti-inflammatory properties. Furthermore, the activity of MPO, an enzyme reflecting the degree of damage and inflammation, was dramatically inhibited by GA. These findings underscore the ability of GA to alleviate colonic inflammation.

The dysfunctional intestinal barrier is a critical factor in the pathogenesis of UC, and UC patients typically exhibit an impaired intestinal barrier [25]. The intestinal epithelial barrier is formed by epithelial cells coupled tightly through tight junction proteins including ZO-1, occludin and claudin-1, which serves as the first physical barrier to maintain intestinal structure and function [35,36]. According to previous studies, tight junction proteins expression was dramatically decreased in the intestinal mucosa of UC patients [37,38]. In this study, we assessed intestinal permeability using FITC-dextran (FD4) as a tracer. Our results suggest that GA significantly reduces intestinal permeability in UC mice. Meanwhile, the expression of tight junction proteins including ZO-1, occludin and claudin-1 was upregulated by GA. We speculate that GA ameliorates UC by restoring the structural and functional integrity of the intestinal barrier, thereby preventing the translocation of harmful bacteria and toxins.

Gut microbiota have been considered as the link between the external environment and the intestinal mucosa [8,39]. A recent study found that dietary supplementation with GA could modulate the imbalance of gut microbiota in mice fed with a high-fat diet [22]. To determine whether gut microbiota serve as the pharmaceutical target for GA in the treatment of UC, we designed co-housing and fecal microbiota transplantation (FMT) experiments. The results confirmed that the gut microbiota modulated by GA primarily contributed to its therapeutic effects on UC.

This study found that GA increased the abundance of Lactobacillus, Oscillospira, Odoribacter and Ruminococcus and decreased the abundance of Mucispirillum, Clostridium, AF12, Sutterella, Allobaculum, Clostridium, Anaerotruncus, Escherichia, Proteus, Klebsiella and Bacteroides. Among these notably changed bacteria, Lactobacillus, a well-known probiotic, was reported to ameliorate UC by maintaining the intestinal barrier and reducing the expression of pro-inflammatory cytokines [40]. Oscillospira, as one of the next-generation probiotic candidates, could produce butyrate and propionate to maintain the integrity of the intestinal barrier [41,42], and it was reported to decrease in abundance in patients with inflammatory bowel disease [43]. Odoribacter, another butyrate producer, is used as the main nutrient by intestinal epithelial cells [44]. Another study found that Odoribacter was sufficient to induce intestinal Th17 cell development and confer resistance against colitis and colorectal cancer [45]. The role of Ruminococcus in UC remains controversial, which shows lower abundance in UC patients than that of healthy individuals [46,47], but under some specific conditions, Ruminococcus promotes the development of colitis [48,49]. Here, the correlation analysis between the micro-environment and gut microbiota revealed that the abundance of Lactobacillus, Oscillospira, Odoribacter, and Ruminococcus abundance was negatively correlated with colonic MPO activity, the DAI score, the spleen index and pro-inflammatory cytokines, while it was positively correlated with colon length, which suggests that GA might ameliorate UC by enriching the abundance of Lactobacillus, Oscillospira, Odoribacter and Ruminococcus. Inversely, the abundance of Escherichia was decreased by GA, which is considered as a pathogenic bacterium that promotes the development of UC [50,51,52]. These findings highlight the ability of GA to restore gut microbiota homeostasis, thereby ameliorating UC.

Microbial metabolites are increasingly recognized as key players in UC pathogenesis [10,11]. Spearman’s correlation analysis was conducted between the differential metabolites with VIP > 3 and notable changed bacteria affected by GA to search for the functional metabolites that have a potentially therapeutic effect on UC. In this study, we screened 4 functional metabolites, which are Gln, GSH, IAAld and DCA.

Gln is the most important energy source for intestinal epithelial cells, promoting mucus synthesis and secretion, and maintaining the intestinal mucus barrier [53]. Glutamine was found to promote the development and proliferation of intestinal epithelial cells [54,55]. Previous studies reported that Gln restriction in cell culture media significantly increased epithelial cell permeability in Caco-2 cells [56]. To confirm the protective effect of Gln on the damage barrier in UC, we constructed a cell damage model of Caco-2 and NCM460 that resembles the intestinal epithelium. Our results showed that Gln could restore the reduction in TEER and the enhancement of paracellular permeability caused by pro-inflammatory cytokines.

Oxidative stress plays an important role in the tissue damage of UC, and excessive oxidative stress may exacerbate intestinal barrier damage and epithelial cell death [57]. GSH, as a type of antioxidant, protects against oxidative stress damage in many tissues [58]. A recent study found that intrinsic GSH plays an important role in promoting mitochondrial function and maintaining intestinal integrity [59]. However, whether GSH plays a role in protecting against intestinal barrier damage in UC is still unknown. In this study, we confirmed that GSH could directly improve barrier dysfunction by maintaining intestinal permeability and associated tight junction proteins in the Caco-2 and NCM460 cell monolayer model.

Numerous tryptophan catabolites from microbiota such as indole-3-ethanol (IEt), indole-3-pyruvate (IPyA) and indole-3-aldehyde (I3A) protect intestinal barrier function [29,60,61,62]. Here, we found that IAAld, an indole derivative of tryptophan catabolites, has a direct protective effect against the dysfunctional barrier. Since Lactobacillus is involved in tryptophan metabolism, we speculate that the upregulation of IAAld may be associated with the upregulation of the abundance of Lactobacillus in this study.

Deoxycholic acid (DCA) is a secondary bile acid that accounts for 58% of bile acids in human feces. It is found that DCA aggravates colonic inflammatory injury and delays the wound healing of colonic epithelial cells [63,64]. We found that DCA exacerbated inflammatory cytokines-induced barrier damage in the Caco-2 and NCM460 cell monolayer model. Collectively, the experimental results suggest GA improves intestinal barrier function by up-regulating the metabolites GSH, Gln and IAAld and down-regulating DCA.

In this study, certain limitations should be acknowledged. Although we identified four functional metabolites through correlation analysis, the direct relationship between these metabolites and GA-modulated gut microbiota requires further investigation. Future studies should explore the causal mechanisms underlying these interactions. Additionally, the translational potential of GA should be evaluated in clinical trials to assess its efficacy and safety in UC patients.

## 4. Materials and Methods

### 4.1. Animals

Male C57BL/6J mice (7~8 weeks old, *n* = 144 for pharmacodynamic experiment, *n* = 72 for co-housing experiment and *n* = 48 for Fecal microbiota transplantation experiment) were purchased from the Animal Center of Peking University Health Center (Beijing, China). The mice were maintained in individually ventilated cage (IVC) systems with a temperature of 25 ± 1 °C under a 12 h light/dark cycle and free access to food and disinfectant water during the whole experimental period. All animal experiments were performed according to the National Institutes of Health Guidelines on the Use of Laboratory Animals. The University Animal Care Committee for Animal Research of Peking University Health Science Center approved the study protocol (Approval NO. LA2022017).

### 4.2. Chemicals and Reagents

The ganoderic acid (GA) was isolated and purified from dried fruiting bodies of G. lucidum as reported previously [65], and the purity of GA was >98%. Dextran sulfate sodium (DSS, molecular weight: 36~50 kDa) was purchased from MP Biochemicals (Santa Ana, CA, USA). The myeloperoxidase (MPO) assay kit was purchased from Nanjing Jiancheng Bioengineering Institute (Nanjing, China). The Elisa assay kits of IL-1β, IL-6, IL-10 and TNF-α were purchased from ABclonal company (Wuhan, China). FD4 was purchased from Sigma-Aldrich (Saint Louis, MO, USA). Rabbit monoclonal ZO-1, occluding and claudin-1 antibodies were purchased from Abcam company (Cambridge, MA, USA). IAAld was purchased from Acmec company (Shanghai, China). DCA, Gln and GSH were purchased from Aladdin company (Shanghai, China)

### 4.3. Establishment and Treatment of Ulcerative Colitis

To evaluate the effect of GA on UC, the mice were randomly divided into six groups after acclimatization for 1 week: the blank control group (Veh group), the DSS-induced UC model group (DSSVeh group), the low-dose GA (16.5 mg/kg)-treated UC group (DSSGA-L group), the middle-dose GA (50 mg/kg)-treated UC group (DSSGA-M group), the high-dose GA (150 mg/kg)-treated UC group (DSSGA-H group) and the sulfasalazine (SASP, 150 mg/kg)-treated UC group (DSSSASP group). The mouse UC model was induced by DSS (2.5% w/vol) in drinking water for 7 days. GA and SASP were dissolved in 0.5% carboxymethylcellulose sodium aqueous solution (vehicle) and administered to the related group mice by oral gavage. Equal volumes of the vehicle were administered to the mice in the control group and model group. GA and SASP were applied for 3 days before inducing UC and maintained until the end of the experiment. The therapeutic experiments were performed by giving GA for 7 days since the third day after the mice were exposed to DSS. The mice were anesthetized by pentobarbital sodium, and colons were obtained for bioanalysis and histopathological examination. The spleens were also removed and weighed. Feces samples were collected for 16S rDNA gene high-throughput sequencing and metabolomics analysis.

### 4.4. Co-Housing Experiment

Male C57BL/6J mice (7~8 weeks old) were randomly divided into six groups: the blank control group (Veh group), the GA-treated control group (GA group), the DSS-induced UC model group (DSSVeh group), the GA-treated UC group (DSSGA group), the co-housing UC model group (Co-DSSVeh group) and the co-housing GA-treated UC group (Co-DSSGA group). The GA dose used in this experiment was 50 mg/kg. The Co-DSSVeh group and Co-DSSGA group mice were raised in the same cage at a 1:1 ratio. UC was induced by DSS (2.5% w/vol) in drinking water for 7 days. GA was applied for 3 days before inducing colitis and maintained until the end of the experiment.

### 4.5. Fecal Microbiota Transplantation

Fecal microbiota transplantation (FMT) was performed as previously reported [14,66]. First, the recipient mice were administered a cocktail of antibiotics (ABX: 1 g/L ampicillin, 0.5 g/L vancomycin, 1 g/L neomycin, 1 g/L metronidazole in sterilized water) for 1 week to deplete the resident microbiota and treated by DSS for another 1 week to induce colitis. Meanwhile, the donor mice were induced by DSS (2.5% w/vol) and received the vehicle or 50 mg/kg GA daily. Then, the antibiotic-treated recipient mice were fed with fresh fecal bacteria collected from vehicle-treated or GA-treated donor mice daily during the period of colitis.

### 4.6. Assessment of the Disease Activity Index

The disease activity index (DAI) was monitored daily, which includes three parameters: body weight loss, stool consistency and gross blood [67]. In brief, the body weight loss score was determined as follows: 0, no weight loss; 1, loss of 1%~5% of original weight; 2, loss of 6%~10% of original weight; 3, loss of 11%~15% of original weight; 4, loss of >15% of original weight. The stool score was determined as follows: 0, well-formed pellets; 1~2, pasty stool that does not stick to the anus; 3~4, liquid stools that adhered to the anus. The bleeding score was determined as follows: 0~1, no blood by using Hemoccult (Beckman Coulter) analysis; 2~3, positive Hemoccult; 4, gross rectal bleeding. The DAI score is the mean value of the score of three parameters.

### 4.7. Histological Assessment

Colons were collected and processed as described previously [33]. The distal colon specimens were fixed in 4% paraformaldehyde (Sigma-Aldrich, Saint Louis, MO, USA) overnight, dehydrated by alcohol (Tong Guang, Beijing, China) and embedded in paraffin. 4 μm thick sections were cut and then stained with hematoxylin (Amresco, Radnor, PA, USA) and eosin (Amresco, Radnor, PA, USA) according to standard protocols. Colon sections were photographed using the Nikon microscope.

### 4.8. Biochemical Analyses

The concentrations of interleukin (IL)- 1β, IL-6, IL-10 and tumor necrosis factor (TNF-α) in colonic tissues were measured by ELISA kits (ABclonal, Wuhan, China) according to the manufacturer’s instructions. The MPO activity in the colonic tissues was measured using a specific kit from Nanjing Jiancheng Bioengineering Institute (Jiancheng, Nanjing, China).

### 4.9. Intestinal Permeability Assay

Fluorescein isothiocyanate (FITC)-dextran was dissolved in neutral phosphate buffer solution (PBS) at a final concentration of 80 mg/mL. Mice were randomly selected from each group, fasted for 4 h and were then given FITC-dextran by gavage at a dose of 600 mg/kg body weight. The mice were anesthetized 3 h later, blood was collected from the orbital venous plexus and serum was collected by centrifugation at 1000× *g* and 4 °C for 12 min. Median fluorescent intensity (MFI) was quantified at 535 nm. The FITC-dextran concentration in the serum was then calculated using a standard curve.

### 4.10. Real-Time qPCR

The total RNA from the colonic tissue was extracted with TRIzol reagent (Takara, Dalian, China) following the manufacturer’s protocol. mRNA was reverse-transcribed into cDNA using an RT-PCR kit (Monad, Wuhan, China). Since DSS inhibits Taq enzyme activity, we added spermine at a final concentration of 0.1 mg/mL to this system to eliminate the inhibitory effect of residual DSS on the Taq enzyme. The cDNA was amplified by real-time PCR using SYBR mix (Monad, Wuhan, China) with an Applied Biosystems^®^ 7500 Real-Time PCR system. The expression of target genes was calculated using the 2^–ΔΔCT^ method and normalized to the expression of Gapdh. The primer sets are listed in Table 1.

### 4.11. Western Blot Analysis

The protein of colonic tissue or cells was extracted using a radioimmunoprecipitation assay buffer (RIPA) containing protease inhibitors and quantified with the bicinchoninic acid regent (Epizyme Biotech, Shanghai, China). The obtained protein was separated by SDS-PAGE, and the discrete protein was transferred onto the PVDF membrane. Subsequently, the membrane was blocked with 5% non-fat milk in TBST at room temperature for 1 h and then incubated at 4 °C overnight with primary antibody ZO-1 (1:1000, Cat #ab221547, Abcam, Cambridge, MA, USA), occludin (1:1000, Cat #ab216327, Abcam) and claudin-1 (1:1000, Cat #ab211737, Abcam). The membrane was washed with TBST three times and incubated with the horseradish peroxidase-conjugated secondary antibody (1:10,000) at room temperature for 2 h. The binding of each antibody was visualized by a super sensitive ECL luminescence reagent (Meilunbio, Dalian, China) according to the manufacturer’s instructions. Protein expression was analyzed by ImageJ version 1.53t software (National Institutes of Health, Bethesda, MD, USA).

### 4.12. Immunofluorescence

For Immunofluorescence, the distal colon specimens were fixed in 4% paraformaldehyde overnight, dehydrated by sucrose (Cat# S112228, Aladdin, Shanghai, China) and embedded in an optimal cutting temperature compound (Cat# 4583, SAKURA, Tokyo, Japan), sectioned at 7 μm thick sections. The sections were blocked with 5% goat serum albumin at room temperature for 1 h and then incubated with the primary antibody ZO-1 (1:100, Cat #ab221547, Abcam), occludin (1:100, Cat #ab215327, Abcam) and claudin-1 (1:100, Cat #ab211737, Abcam) at 4 °C for 12 h. Subsequently, the sections were washed with cold PBS 3 times and incubated with the FITC-conjugated secondary antibody at room temperature for 2 h in the dark. Finally, the sections were counterstained with DAPI (Cat# D9542, Sigma, Saint Louis, MO, USA). Images were captured with a Nikon fluorescence microscope (Nikon, Tokyo, Japan, ECLIPSE Ti2-U).

### 4.13. 16S rDNA High-Throughput Sequencing and Analysis

Fresh feces was collected and frozen at −80 °C within 30 min. The total DNA of the fecal microbiota was extracted by a QIAquick Gel Extraction Kit (Cat# 28706X4, Qiagen, Hilden, Germany). Universal primers (B341F, 5′-CCTACGGGNGGCWGCAG-3′) and (B785R, 5′-GACTACHVGGGTATCTAATCC-3′) were used for amplification of the V3-V4 hypervariable regions of the prokaryotic 16S rDNA gene. Amplification products were detected by 2% agarose gel electrophoresis, recovered and purified with Agencourt AMPure XP kit (Cat# A63880, Beckman Coulter, Inc., Brea, CA, USA). Qualified libraries performed high-throughput sequencing using the Illumina MiSeq platform (Illumina, Inc., San Diego, CA, USA) to obtain raw sequence data. After the original sequence information was dismounted, the low-quality data were deleted through splicing and filtering, and high-quality analysis sequences were obtained. Clean reads were clustered by a 97% similarity standard to obtain operational taxonomic units (OTUs), each of which is considered to represent a species. According to OTU abundance, species abundance and cluster analysis, intra-sample diversity (α-diversity) analysis, inter-sample diversity (β-diversity) analysis and inter-group difference analysis were calculated by QIIME (quantitative insights into microbial ecology, http://www.qiime.org (accessed on 19 August 2021)).

### 4.14. UHPLC-MS/MS Analysis

The samples were pre-cooled with 900 μL of methanol/acetonitrile/water (2:2:1, *V*/*V*/*V*). An ultrasound was performed in an ice bath for more than 60 min and the samples were incubated at −20 °C for 1 h to precipitate protein. Subsequently, the samples were centrifuged at 16000 g at 4 °C for 20 min. The supernatants were recovered for vacuum drying and then re-dissolved with 100 μL of acetonitrile–water solution (1:1, *V*/*V*). Next, the mixtures were centrifuged at 16,000× *g* at 4 °C for 15 min and the supernatant was injected for UHPLC-MS/MS analysis. Metabolomics profiling was analyzed using a UPLC-ESI-Q-Orbitrap-MS system (UHPLC, Shimadzu Nexera X2 LC-30AD, Shimadzu, Japan) coupled with Q-Exactive Plus (Thermo Scientific, San Jose, USA). Data preprocessing and multivariate statistical analysis were performed as reported previously [68,69].

### 4.15. Cell Culture and Metabolites Treatment

Human colonic adenocarcinoma cell line Caco-2 and human normal colonic epithelial cell line NCM460 were maintained at 37 °C and under 5% CO_2_ in DMEM supplemented with 10% FBS. The medium was replaced every 2 days. For metabolites treatment, the cells were seeded in six-well plates at a density of 8 × 10^5^ cells/well and treated with 100 μM DCA, IAAld, Gln and GSH for 12 h. Subsequently, 10 ng/mL TNF-α + IFN-γ was added to the plates to induce cell damage. After 24 h, the cells were harvested for quantifying the expression of tight junction proteins.

### 4.16. Measurement of Transepithelial Electrical Resistance

Caco-2 cells were planted on the transwell (1 × 10^5^ cells per well, 1.12 cm^2^ surface area of filter membrane) for 2 weeks to form the IEC monolayers. The medium was replaced every 2 days. Subsequently, the cell monolayers were treated with 100 μM DCA, IAAld, Gln and GSH for 12 h and then 10 ng/mL TNF-α + IFN-γ was added to induce the damage for 24 h. Transepithelial electrical resistance (TEER) was measured using a Millicell-ERS volt-ohmmeter (Cat# EVOM2, Millipore, Bedford, MA) before and after the induction of barrier damage. The loss of TEER in the treated groups was normalized to that in the control group.

### 4.17. Paracellular Permeability Assay

The paracellular permeability of Caco-2 cells monolayers was measured by Lucifer yellow (Cat# L131282, Sigma-Aldrich, Shanghai, China) as the paracellular transport marker. Briefly, the monolayers were washed with pre-warmed PBS and 0.5 mL PBS containing 100 μg/mL Lucifer yellow added to the apical compartment (*V_Apical_*) and 1.5 mL PBS added to the basolateral compartment (*V_Basolateral_*) of the transwell described in 4.16. After 30 min, the intensities of fluorescence in the apical (*RFU_Apica_*_l_) and basolateral (*RFU_Basolateral_*) compartment were determined by the apical compartment at excitation and emission wavelengths of 485 and 535 nm. Permeability was calculated according to the following formula:$$\%Lucifer\;Yellow=\frac{V_{Basolateral}\;*\;{RFU}_{Basolateral}}{V_{Apical}\;*\;{RFU}_{Apical}+V_{Basolateral}\;*\;{RFU}_{Basolateral}}$$

The permeability of the treated groups was normalized to that of the control group.

### 4.18. Statistical Analysis

All data are expressed as the mean ± SEM. Statistical analyses were performed using GraphPad Prism 5.0 software (GraphPad Software, Inc., San Diego, CA, USA). For multiple comparisons, an analysis of metabolite content was performed using Student’s *t*-test. Animal experiments and microbiota were performed using one-way ANOVA. *p* < 0.05 was considered statistically significant.

## 5. Conclusions

In this study, we determined the effect of GA on UC using the UC mouse model induced by dextran sulfate sodium (DSS). The experimental results showed that GA could alleviate histopathological damage, inflammation and intestinal barrier disruption in the mouse colon, in which gut microbiota played a critical role for the pharmacological activities of GA. The experimental results indicate that gut microbiota-related metabolites affected by GA had a protective effect on the intestinal barrier function by regulating the expression of tight junction proteins (Figure 8). All these data suggest that GA might be developed as a potential pharmaceutical candidate for UC treatment.

## Figures and Tables

**Figure 1 ijms-26-02466-f001:**
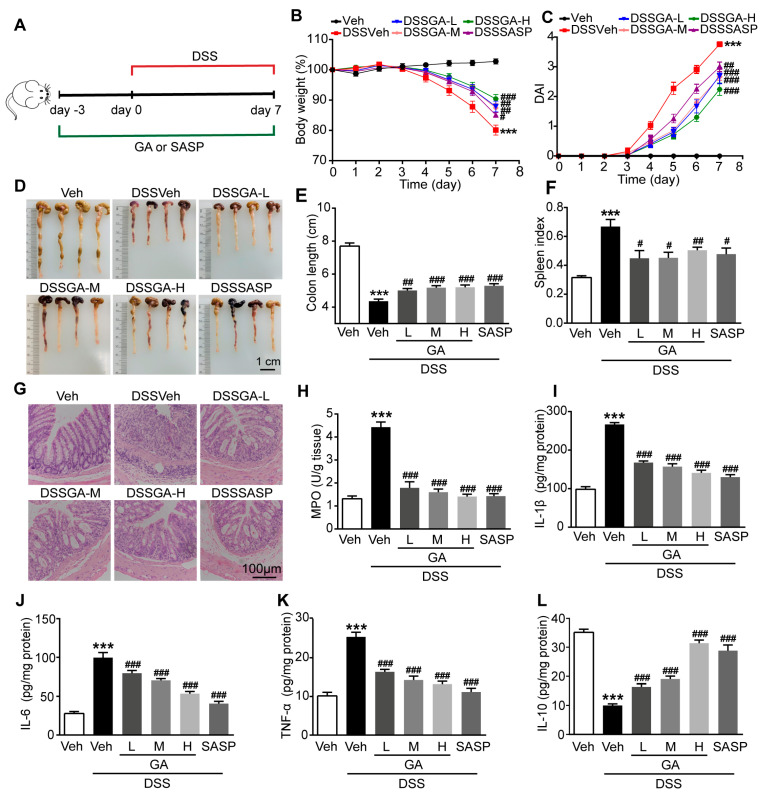
Effect of GA on pathological symptoms and colonic inflammation in DSS-induced UC mice. (**A**) Experimental procedure for studying the effect of GA on DSS-induced UC in mice. (**B**) Daily changes in body weight. (**C**) DAI. (**D**) Colon appearance, Scale bar = 1 cm. (**E**) Colon length. (**F**) Spleen index. (**G**) Representative pictures of H&E-stained colon tissue (magnification of ×100). Scale bar = 100 µm. (**H**) MPO activity. (**I**) IL-1β level. (**J**) IL-6 level. (**K**) TNF-α level. (**L**) IL-10 level. Values are shown as the mean ± SEM (*n* = 12). *** *p* < 0.001 vs. Veh group. # *p* < 0.05, ## *p* < 0.01, ### *p* < 0.001 vs. DSSVeh group.

**Figure 2 ijms-26-02466-f002:**
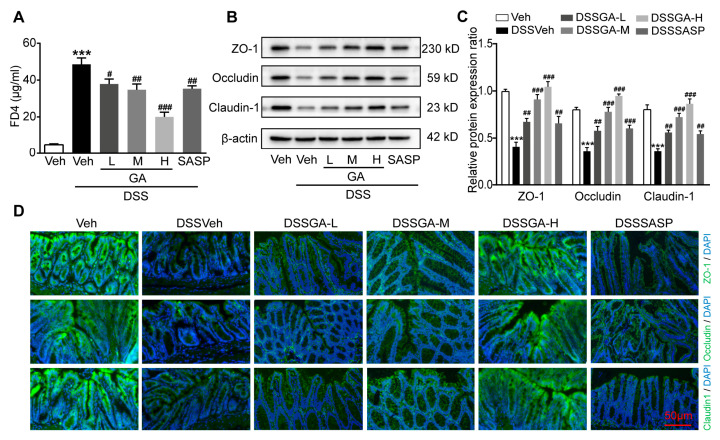
Effect of GA on intestinal barrier function in DSS-induced UC mice. (**A**) Levels of FD-4 in blood. (**B**) Representative Western blots of tight junction markers. (**C**) Relative expression levels of ZO-1, occludin and claudin-1 proteins. (**D**) Representative ZO-1, occludin and claudin-1 immunofluorescence staining of colon tissues (magnification of ×200). Scale bar = 50 µm. Values are shown as the mean ± SEM (*n* = 5). *** *p* < 0.001 vs. Veh group. # *p* < 0.05, ## *p* < 0.01, ### *p* < 0.001 vs. DSSVeh group.

**Figure 3 ijms-26-02466-f003:**
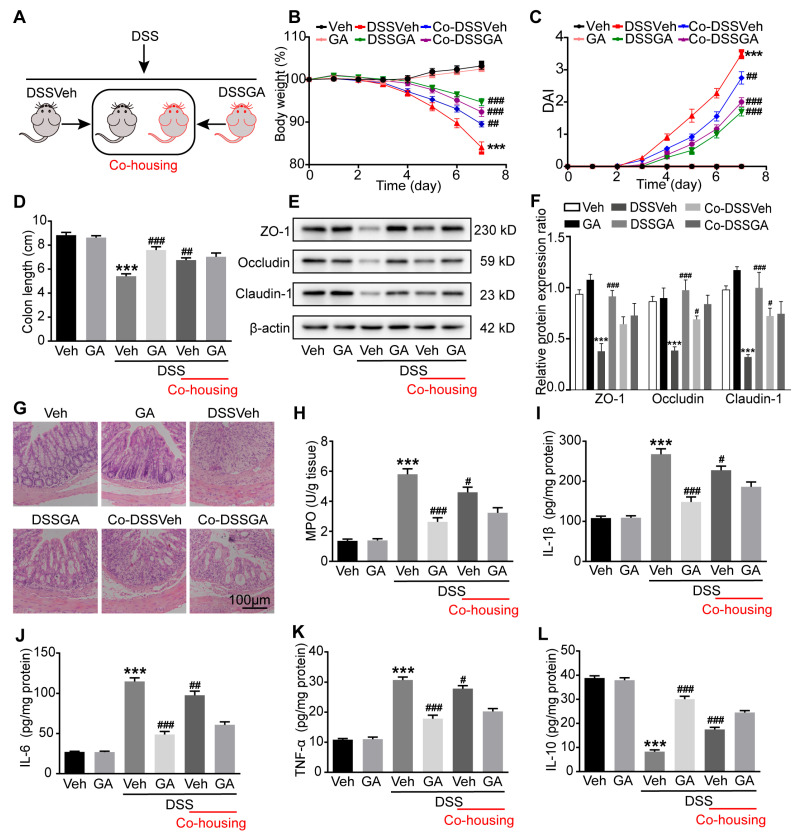
Effect of co-housing on anti-UC activity of GA. (**A**) Experimental procedure of co-housing. (**B**) Daily changes in body weight. (**C**) DAI. (**D**) Colon length. (**E**) Representative Western blots of tight junction markers. (**F**) Relative expression levels of ZO-1, occludin and claudin-1 proteins. (**G**) Representative pictures of H&E-stained colon tissue (magnification of ×100); Scale bar = 100 µm. (**H**) MPO activity. (**I**) IL-1β level. (**J**) IL-6 level. (**K**) TNF-α level. (**L**) IL-10 level. Values are shown as the mean ± SEM (*n* = 12 for (**A**–**D**) and (**H**–**L**), *n* = 5 for (**F**)). *** *p* < 0.001 vs. Veh group. # *p* < 0.05, ## *p* < 0.01, ### *p* < 0.001 vs. DSSVeh group.

**Figure 4 ijms-26-02466-f004:**
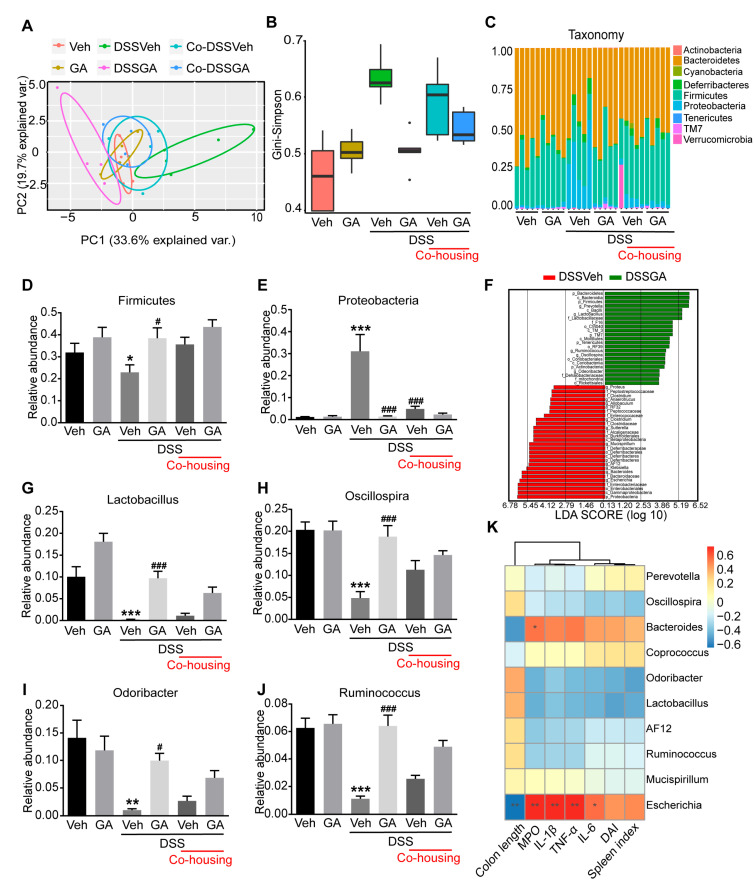
Effect of GA on gut microbiota dysbiosis in DSS-induced UC mice. (**A**) Multiple sample PCoA of the Bray–Curtis distance based on OTUs. (**B**) Microbial α diversity (Higher Gini–Simpson index indicates lower α diversity). (**C**) Bar chart of the bacterial community composition at the phylum level. (**D**) Relative abundance of Firmicutes. (**E**) Relative abundance of Proteobacteria. (**F**) LEfSe analysis between the DSSVeh group and DSSGA group. (**G**) Relative abundance of Lactobacillus. (**H**) Relative abundance of Oscillospira. (**I**) Relative abundance of Odoribacter. (**J**) Relative abundance of Ruminococcus. (**K**) Correlation analysis between the 10 most dominant genera in all samples and micro-environmental factors. Values are shown as the mean ± SEM (*n* = 5). * *p* < 0.05, ** *p* < 0.01, *** *p* < 0.001 vs. Veh group. # *p* < 0.05, ### *p* < 0.001 vs. DSSVeh group.

**Figure 5 ijms-26-02466-f005:**
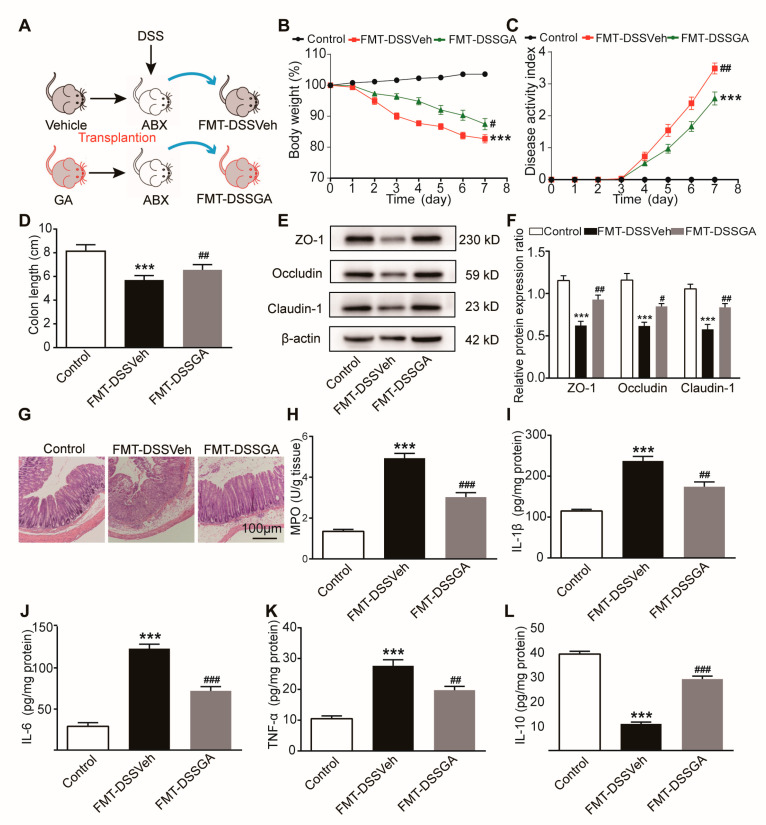
Effect of FMT on anti-UC activity of GA. (**A**) Experimental procedure of ABX-mediated gut microbiota depletion and FMT. (**B**) Daily changes in body weight. (**C**) DAI. (**D**) Colon length. (**E**) Representative Western blots of tight junction markers. (**F**) Relative levels of ZO-1, occludin and claudin-1 proteins. (**G**) Representative pictures of H&E-stained colon tissue (magnification of ×100). Scale bar = 100 µm. (**H**) MPO activity, (**I**) Relative level of IL-1β. (**J**) Relative level of IL-6. (**K**) Relative level of TNF-α. (**L**) Relative level of IL-10. Values are shown as the mean ± SEM (*n* = 12 for (**A**–**D**) and (**H**–**L**), *n* = 5 for (**F**)). *** *p* < 0.001 vs. Control group. # *p* < 0.05, ## *p* < 0.01, ### *p* < 0.001 vs. FMT-DSSVeh group.

**Figure 6 ijms-26-02466-f006:**
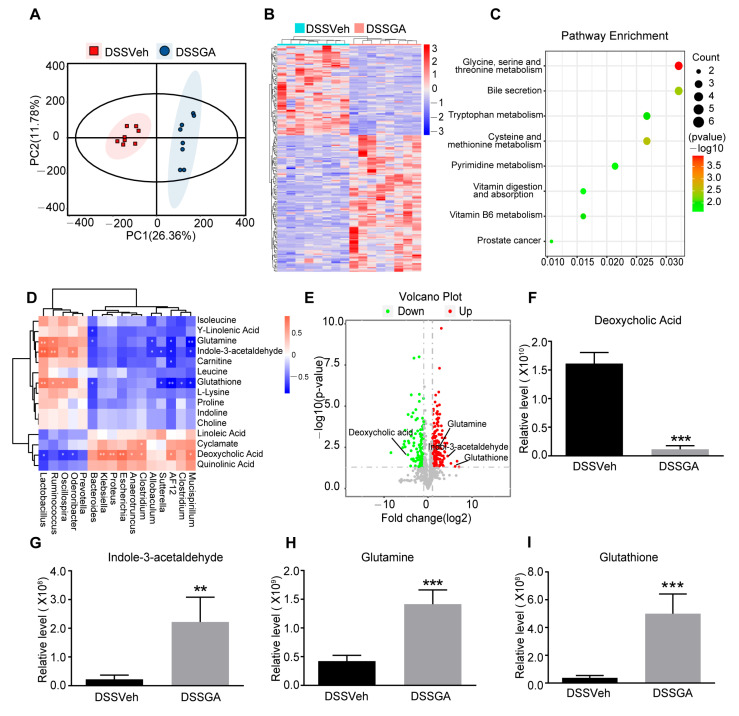
Key differential metabolites affected by GA with untargeted metabolomics assay. (**A**) Partial least squares discrimination analysis (PLS-DA). (**B**) Heatmap overview of differential metabolites in feces. (**C**) Pathway enrichment analysis based on KEGG database. (**D**) Spearman’s correlation analysis between differential metabolites with VIP > 3 and notable changed bacteria affected by GA. (**E**) Volcano plot representing differential metabolites. (**F**) Relative level of DCA. (**G**) Relative level of IAAld. (**H**) Relative level of Gln. (**I**) Relative level of GSH. Values are shown as the mean ± SEM (*n* = 8). ** *p* < 0.01, *** *p* < 0.001 vs. DSSVeh group.

**Figure 7 ijms-26-02466-f007:**
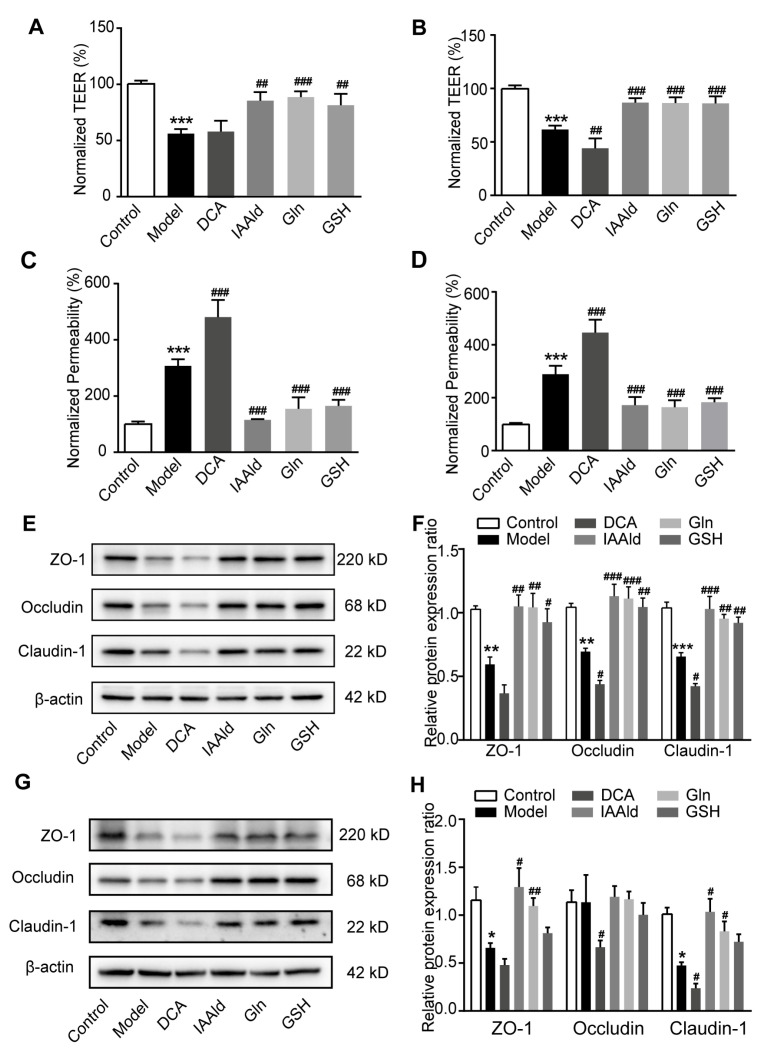
Effect of IAAld, Gln and GSH on intestinal barrier damage. (**A**) Transepithelial electrical resistance (TEER) of polarized Caco-2 monolayers. (**B**) TEER of polarized NCM460 monolayers. (**C**) Paracellular permeability of Caco-2 cells monolayers. (**D**) Paracellular permeability of NCM460 cells monolayers. (**E**) Representative Western blots of tight junction markers in the Caco-2 cell. (**F**) Relative expression levels of ZO-1, occludin and claudin-1 proteins in the Caco-2 cell. (**G**) Representative Western blots of tight junction markers in the NCM460 cell. (**H**) Relative expression levels of ZO-1, occludin and claudin-1 proteins in the NCM460 cell. The TEER and permeability of treated groups were normalized to that of the control group. Values are shown as the mean ± SEM (*n* = 5). * *p* < 0.05, ** *p* < 0.01, *** *p* < 0.001 vs. control group. # *p* < 0.05, ## *p* < 0.01, ### *p* < 0.001 vs. model group.

**Figure 8 ijms-26-02466-f008:**
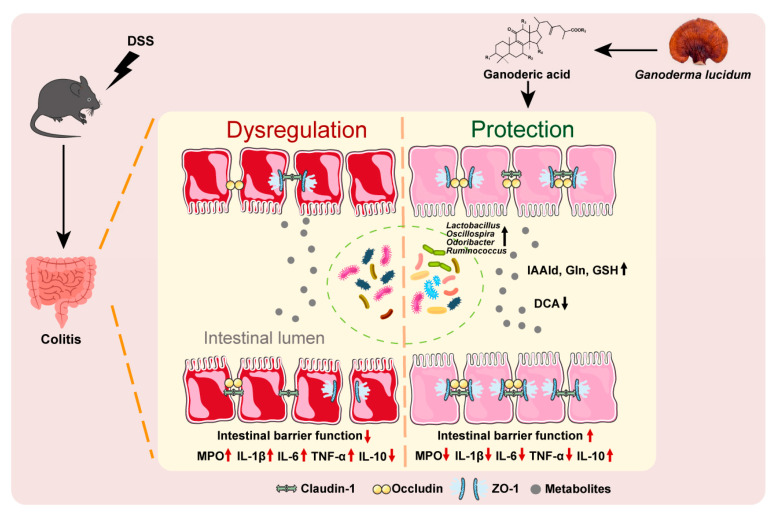
Schematic diagram of the proposed mechanisms underlying the effect of GA in DSS-induced UC mice.

**Table 1 ijms-26-02466-t001:** Sequences of the gene-specific primers.

Gene	FORWARD PRIMER (5′-3′)	REVERSE PRIMER (5′-3′)
IL-1β	TCCATGAGCTTTGTACAAGGA	AGCCCATACTTTAGGAAGACA
IL-6	GTTCTCTGGGAAATCGTGGA	TGTACTCCAGGTAGCTA
TNF-α	AGACCCTCACACTCAGATCA	TCTTTGAGATCCATGCCGTTG
IL-10	ATTTGAATTCCCTGGGTGAGAAG	CACAGGGGAGAAATCGATGACA
GAPDH	AGGTCGGTGTGAACGGATTTG	TGTAGACCATGTAGTTGAGGTCA

## Data Availability

The data are available on request from the authors.

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
