# Peer review of "Ganoderic Acid Ameliorates Ulcerative Colitis by Improving Intestinal Barrier Function via Gut Microbiota Modulation"

_ijms, 2025, doi:10.3390/ijms26062466_

Round 1

Reviewer 1 Report

Comments and Suggestions for Authors

The authors have conducted an interesting study on the effects of ganoderic acid on an ulcerative colitis animal model. Please consider the following comments and reply to each of them indicating the number of lines in the new version of the manuscript that contains the modifications.

Comment 1: Please double-check that all acronyms are defined the first time they are used. This applies separately to the abstract and the rest of the manuscript. For example, IAAld in lines 31 and 236.

Comment 2: Please note that many acronyms are defined in the Materials and Methods section. However, this section shows up after Results and Discussion. Please move all definitions to the first spot where they appear, regardless if it is the Results or Discussion sections.

Comment 3: Please mode the text in lines 64 to 70 to Conclusions or Discussion, as convenient.

Comment 4: Please adapt the statement in the “Data availability statement” section in line 579, as the raw data is not available in the article or in the supplementary material (only a supplementary figure is available).

Comment 5: Please indicate the number of mice in line 378.

Comment 6: Please clarify in which cases Student t-test or ANOVA were used (lines 554-555).

Comment 7: In the captions of Figure 1, use capital letter for “A-D” in line 105.

Comment 8: Please re-design the setting of images in Figures so that they can actually be understood. Being that small will prevent readers ability to understand the details. Particularly Figures 1B, 1C, 2C, 3B, 3C, 3F, 4C, 4F, 4K, 5F, 6B, 6C, 6D, 6E, 7F and 7H  must be presented in a bigger size.

Comment 9: In line 482, indicate the company and location for ImageJ.

Author Response

Comments 1: Please double-check that all acronyms are defined the first time they are used. This applies separately to the abstract and the rest of the manuscript. For example, IAAld in lines 31 and 236.

Response: Thank you for pointing this out. In accordance with the recommendations, we have made corresponding additions and revisions to lines 27, 34, 98, 99, 100, 101, 125, 185, 186, 212, 245, 246, 247, 249 and 251 of the revised manuscript

Comments 2: Please note that many acronyms are defined in the Materials and Methods section. However, this section shows up after Results and Discussion. Please move all definitions to the first spot where they appear, regardless if it is the Results or Discussion sections.

Response: In accordance with the recommendations, we have implemented modifications to lines 412, 413, 415, and 416 of the revised manuscript.

Comments 3: Please mode the text in lines 64 to 70 to Conclusions or Discussion, as convenient.

Response: This section has been relocated to the conclusion of the article and can now be found in lines 583-586 of the revised manuscript.

Comments 4: Please adapt the statement in the “Data availability statement” section in line 579, as the raw data is not available in the article or in the supplementary material (only a supplementary figure is available).

Response: The revisions have been implemented as advised and can be found in line 605 of the revised manuscript.

Comments 5: Please indicate the number of mice in line 378.

Response: Additional descriptions have been incorporated into the revised manuscript at lines 398-399.

Comments 6: Please clarify in which cases Student t-test or ANOVA were used (lines 554-555).

Response: The manuscript has been revised in accordance with the recommendations, with pertinent information added to lines 579-581 of the revised manuscript.

Comments 7: In the captions of Figure 1, use capital letter for “A-D” in line 105.

Response: Upon meticulous review, we regret to acknowledge an error in our previous notation. Additionally, corresponding amendments have been made to lines 115, 166, and 229 of the revised manuscript.

Comments 8: Please re-design the setting of images in Figures so that they can actually be understood. Being that small will prevent readers ability to understand the details. Particularly Figures 1B, 1C, 2C, 3B, 3C, 3F, 4C, 4F, 4K, 5F, 6B, 6C, 6D, 6E, 7F and 7H must be presented in a bigger size.

Response: We have adjusted the size of our figures as per the recommendations. We have identified that the image quality may have been compromised during the manuscript upload process. To address this, we will provide the editorial office with a PDF version of the manuscript and a file containing all the figures for clarity and quality assurance.

Comments 9: In line 482, indicate the company and location for ImageJ.

Response: Supplementary information has been added to lines 505-506 of the revised manuscript in accordance with the suggestions provided.

Thank you for your valuable comments and suggestions. Attached is our revised manuscript.

Reviewer 2 Report

Comments and Suggestions for Authors

The manuscript entitled “Ganoderic acid ameliorates ulcerative colitis by improving intestinal barrier function via gut microbiota modulation” is a document that has recent, very interesting information and the subject matter is novel. They address studies of the main bioactive compound found in Ganoderma lucidum and how it has a positive effect against ulcerative colitis and also how intestinal functions are improved. They have a good experimental design. However, the information they present in the introduction is too little and generalized to introduce the reader to the research topic, I suggest that they improve it. Above all, in exposing the main effects of ulcerative colitis; talk a little more about the properties of Ganoderic acid, its mechanism, but above all expand the information on its source (Ganoderma lucidum). I consider that they should also expand the discussions a little more, they have many interesting results and applied many novel methodologies that could better explain the results obtained. Another detail that I consider important for you to take into account is the visual quality of the figures. They are very small and when you enlarge them they become distorted and the information cannot be properly appreciated. In addition, some specific details that I point out below:
L64-L70: This information should be in summary since it is the results and conclusion of this investigation and should not be part of the introduction. Please change it.
L287-288: This paragraph should not be in this section since it is a conclusion. I suggest you remove it.
Section 4.1 I consider it important that you declare whether you obtained the endorsement or permission of a bioethics committee that certifies the good handling of animals.
L547: you must describe what type of equipment you used to perform the test.
L549: You must describe the meaning of the abbreviations in the formula.

Author Response

Comments 1: L64-L70: This information should be in summary since it is the results and conclusion of this investigation and should not be part of the introduction. Please change it.

Response: Thank you for pointing this out. This section has been relocated to the conclusion of the article and can now be found in lines 583-586 of the revised manuscript.

Comments 2: L287-288: This paragraph should not be in this section since it is a conclusion. I suggest you remove it.

Response: The revisions have been made in accordance with the suggestions provided.

Comments 3: Section 4.1 I consider it important that you declare whether you obtained the endorsement or permission of a bioethics committee that certifies the good handling of animals.

Response: In accordance with the recommendations, pertinent information has been added to lines 403-406 of the revised manuscript.

Comments 4: L547: you must describe what type of equipment you used to perform the test.

Response: In accordance with the recommendations, information regarding the type of equipment has been supplemented at line 569 of the revised manuscript.

Comments 5: L549: You must describe the meaning of the abbreviations in the formula.

Response: In line with the recommendations, additional relevant information has been incorporated into lines 568-570 of the revised manuscript.

In addition, we have revised and improved the Introduction and Discussion sections of the manuscript based on your suggestions.

Thank you for your valuable comments and suggestions. Attached is our revised manuscript.

Round 2

Reviewer 1 Report

Comments and Suggestions for Authors

Good job.

Please add all figures in the highest resolution possible as supplementary material so that readers can access them and see the details that will not be visible in the PDF paper.

Author Response

Comments 1: Please add all figures in the highest resolution possible as supplementary material so that readers can access them and see the details that will not be visible in the PDF paper.

Response: Thank you for your suggestion. We have packaged all figures in the highest resolution into a single file and uploaded it to the “Figures, Graphics, Images” option in the submission system.